# The Tribological Properties of 30CrMnSiA Bearing Steels Treated by the Strengthening Grinding Process under Lubrication Wear

**DOI:** 10.3390/ma15207380

**Published:** 2022-10-21

**Authors:** Xiaochu Liu, Xiujie Chen, Zhongwei Liang, Tao Zou, Zhaoyang Liu, Jinrui Xiao, Dongwei Li, Diaodiao Yu

**Affiliations:** 1School of Mechanical and Electrical Engineering, Guangzhou University, Guangzhou 510006, China; 2Guangzhou Key Laboratory of Strengthen Grinding and High-Performance Machining, Guangzhou University, Guangzhou 510006, China; 3Guangdong Research Centre for Strengthen Grinding and High-Performance Micro/Nano Machining, Guangzhou 510006, China

**Keywords:** 30CrMnSiA steel, strengthening grinding process, tribological properties, lubrication

## Abstract

This study used the strengthening grinding process (SGP) to treat the surface of 30CrMnSiA bearing steels. The effect of the jet angle of SGP on the tribological properties of 30CrMnSiA bearing steels under lubrication was investigated. The principle of enhancing wear resistance of 30CrMnSiA bearing steel ascribed to SGP was discussed in detail. The results showed that the lubrication properties and surface hardness of the 30CrMnSiA steels were enhanced due to the formation of numerous microscale microscope oil pockets on the surface layer and the grain refinement of the surface microstructures, resulting in a significant improvement in wear resistance. With the jet angle of SGP increased from 0° to 90°, the friction coefficient, the wear volume, and the specific wear rate were exhibited to reduce rapidly first, then reduce slowly, and then rise slowly. With the optimal parameters at the jet angle of 60°, compared with the control sample, the average friction coefficient was reduced from 0.2235 to 0.1609, and the wear volume and specific wear rate were reduced from 9.04 × 10^−3^ mm^3^ to 3.82 × 10^−3^ mm^3^ and from 15.13 × 10^−3^ mm^2^/N to 6.36 × 10^−3^ mm^2^/N, respectively. When the jet angle was 90°, the reduced wear resistance was mainly attributed to the excessive roughness that caused the oil coating on the surface to be severely damaged.

## 1. Introduction

30CrMnSiA medium carbon steel has been widely utilized in equipment manufacturing and aerospace industries, such as propellers, engines pistons, and motor bearings, owing to its excellent mechanical properties, such as high strength, superior toughness, and hardenability [1,2,3,4,5]. For 30CrMnSiA steel in its original state, its poor tribological properties limit its use in practical engineering applications. It is well known that tribological properties are essential factors affecting the stability and reliability of workpieces [6]. Wear of the workpiece can lead to many problems, such as a decrease in accuracy, surface condition, and stability. Severe wear even leads to alterations in the surface structure of the material and mechanical failure [7,8], which is mainly due to the rupture of the oil film and lubricant failure, resulting in dry friction [9,10]. Thus, it is necessary to improve the wear resistance properties of the contact surfaces to prevent the occurrence of dry frictional wear.

To address this issue, many methods have been carried out to enhance the tribological properties of the metallic workpiece, such as surface coating, surface chemical heat treatment, and surface mechanical strengthening. Wang et al. [11] used an unbalanced magnetron sputtering technique for the surface of M2 steel to obtain Cr(N)/C(DLC) multilayered coatings deposited and investigated the wear properties of the different thicknesses of the coatings under dry and wet sliding. The research found that the specimens with a thinner coating exhibited a lower tendency to form fatigue cracks under dry slide wear. Furthermore, the thinnest coating, which had the best anti-wear performance in the dry testing condition, exhibited the worst wear resistance in wateriness. Luo et al. [12] combined diffusion reactions with surface coating techniques and annealed the Ti-6AI-4V alloys covered with an Ni/Cu/Ni coating to enhance their tribological properties. It was demonstrated that reinforcing phases with high hardness and strength were produced in the Ni/Cu/Ni coating and improved the microhardness, reduced the friction coefficient, and lessened the wear rate. Yan et al. [13] used the laser quenching technique (LQ) as a subsequent procedure of the typical plasma nitriding treatment process (PN) to improve the surface properties of 30CrMnSiA steel. The microstructure and properties of such layers were compared with those obtained by PN or LQ treatment. The study showed that the specimen by PN+LQ treatment exhibited better wear resistance. This was attributed to the formation of retained austenite and Fe_3_O_4_ in the modified layer of 30CrMnSiA steel, which contributed to the improved impact toughness and lubrication during sliding. Nevertheless, the severe pitting problem occurred once the coating had been damaged, and the thinner modified layer formed after the surface chemical heat treatment was vulnerable to surface cracking. It is also for these reasons that their applications are limited in engineering. As an alternative, the surface mechanical strengthening technique has been a successful solution for obtaining excellent wear resistance of metals. Han et al. [14] investigated the tribological behavior of different austempered AISI 5160 steel specimens with shot peening in order to enhance workpiece performance. The study indicated that the shot peening produced higher hardness on soft austempered specimens rather than on hard austempered specimens. Compared with non-shot peened austempered disk specimens, the wear reduction of disk specimens with shot peening was up to 73%. The influences of the ultrasonic surface rolling process (USRP) on the surface integrity and wear resistance of 300M steel samples were comprehensively studied by Dang et al. [15]. The results illustrated that nanocrystalline and ultrafine crystalline structures were produced in the surface layer by USRP treatment. The comprehensive effects of grain refinement, work hardening, and compressive residual stress improved the surface hardness by about 30.9%, which enhanced the wear resistance of 300M steel. However, the traditional shot peening process generally produces high roughness on the workpiece surface, and the protection ability is still far from practical demands [16]. On the other hand, surface rolling creates a hardened layer with an apparent delamination phenomenon of the internal material, which easily leads to the surface layer peeling off [17]. With the rapid development of the mechanical industry, new techniques to improve the anti-corrosion properties of steel are urgently needed.

As a new surface mechanical strengthening technique, the strengthening grinding process (SGP) has been used widely to improve the surface hardness, wear resistance, and impact resistance for various metallic materials such as bearing-, carbon-, alloy-, and stainless-steel, as well as carbon irons [18,19,20,21]. The basic principle of SGP is that under the impetus of compressed gas, the steel beads covered with strengthening liquids and abrasive powder will impact the surface layer on the workpiece to produce a microcutting effect to remove the deterioration layer and significantly improve the surface properties of the workpiece. This is a solid–liquid–gas three phase coupled process that creates a thicker strengthening layer on the surface of metallic materials [22], which remarkably improves the surface properties of metallic materials. The SGP has been proposed and engaged in research since 2007 by Liu [23]. During the past decade, the introduction of SGP in metallic materials to improve surface properties has been reported [24,25,26]. Metallic materials obtain a double composition layer that consists of the surface-modified layer and the near-surface-strengthened layer by SGP [27]. Moreover, the technique can enhance the surface mechanical properties of materials, such as roughness, corrosion, wear resistance, and impact resistance by producing grain refinement and micropit morphology on the surface layer [28,29]. Therefore, the SGP is an effective surface strengthening and modification method and can show perfect universality for many kinds of metallic materials.

In this work, the 30CrMnSiA steel sample was post-treated by SGP. Through the tribological experiments under lubrication, the effect of the jet angle of SGP on the wear behavior of samples was analyzed. The effect of microstructure and matrix properties on the wear behavior of 30CrMnSiA steel was studied. The principle of SGP-reduced wear resistance was discussed to provide experimental support for the applications of SGP in the field of aerospace tribology.

## 2. Materials and Methods

### 2.1. Materials

The experiment material was a commercial 30CrMnSiA medium-carbon steel. An energy dispersive spectrometer (EDS, Oxford X-Max 50, Oxford, UK) was used to test the chemical composition of 30CrMnSiA, and the result is shown in Table 1. The material was austenitic at 880 °C for 45 min, and then tempered at 540 °C for 180 min after oil cooling and quenching, and finally removed and air-cooled to the room temperature of 30 °C. The heat-treated steel was processed into SGP samples of 10 mm × 10 mm × 5 mm by means of electrical discharge machining wire cutting. The surfaces of the samples were ground and polished with #180–1500 sandpaper to ensure the surface roughness *R*_a_ was below 1.6 um. The microstructure of 30CrMnSiA steel was observed using a scanning electron microscope (SEM, TESCAN MIRA4, TESCAN Inc., Brno, Czech Republic) and was found to have a mainly lath martensitic and retained austenitic structure, as shown in Figure 1.

### 2.2. Strengthening Grinding Process and Surface Treatment of Samples

The tested samples were surface-treated by homemade strengthening grinding processing equipment, and the working principle of the SGP equipment is schematically illustrated in Figure 2. An air compressor generates the compressed gas, and the air pressure is set at 0.7 MPa. The abrasive materials are composed of millimeter-scale steel beads, micrometer-scale abrasive powders, and the strengthening liquid. The initial kinetic energy produced by the compressed gas was delivered to the treatment of the surface by the abrasive materials. During the procedure, a three-phase mixed jet flow comprised the steel beads, the abrasive powder, the strengthening liquid, and the compressed gas. The G10-bearing balls with a particle size of 0.8 mm were selected as steel beads. The alumina powder with a particle size of 1.3 µm and an average Vickers hardness of 2.2 × 10^5^ MPa was used as the abrasive powder. The strengthening liquid was composed of borax (i.e., Na_2_B_4_O_7_·10H_2_O), triethanolamine (C_6_H_15_NO_3_), benzotriazole (C_6_H_5_N_3_), and distilled water [30]. The jetting distance of abrasive materials was set as 90 mm, and the processing time was 15 min.

The working parameters of SGP are presented in Table 2. One sample without SGP treatment was set as a control group. Three samples were treated with the jet angles at 30°, 60°, and 90°, respectively. The samples were ultrasonically cleaned with 95% alcohol and dried before being treated with the SGP.

### 2.3. Friction and Wear Testing

A homemade reciprocating friction and wear tester was used to study the oil lubrication sliding friction and wear behavior of four samples at 30 °C room temperature. Figure 3 shows a schematic diagram of the experimental principle. Si_3_N_4_ ceramic balls with a diameter of 8 mm were used against wear parts, and their hardness ranged from 65 to 67 HRC. The experimental parameters were set as follows: the load was 20 N, the amplitude was 3 mm, the frequency was 6 Hz, and the running time was 3600 s. Before and after each experiment, the samples were ultrasonically cleaned with 95% alcohol and dried treated.

### 2.4. Characterization

After the SGP treatment, the surface morphology of the samples was observed by SEM, and surface profiles were analyzed using a white light interferometer (Rtec UP-3000, Rtec instruments Inc., Silicon Valley, CA, USA).

The metallographic specimens were cut from the cross sections of four samples, and the hard profiles of their cross-section were measured with an microhardness tester (HV-1000, Shanghai Wanheng Precision Instruments Co., Ltd., Shanghai, China). A diamond indenter was used for the test, and the test load and load time were set at 200 g and 15 s, respectively. Five indentations were placed at intervals of 100 µm on the cross-section, and the average was used to prepare the hardness profile.

The phase compositions of tested samples were identified by an X-ray diffractometer (Rigaku Smartlab 9KW, Rigaku Corp, Akishima, Japan). The test surface was tested by Cu-Kα radiation in the glancing angle range of 20°–90° and recorded with a 0.5° interval step at 40 kV and 30 mA.

The friction coefficient was monitored by the matching program of the testing machine. The profile and volume of wear marks were measured by a white light interferometer. Furthermore, the calculation formulas of wear volume and specific wear rate were as follows [31]:(1)$$W_{V}=W_{D}\cdot W_{W}\cdot S$$
(2)$$W_{R}=W_{V}/\left(F_{H}\cdot S\right)$$
where *W*_V_ is the wear volume, *W*_D_ is the average depth of the wear mark, *S* is the length of the wear mark, *W*_R_ is the specific wear rate, and *F*_H_ is the load. The microscopic wear morphology was observed by SEM.

## 3. Results

### 3.1. Surface Morphology

The SEM maps, three-dimensional morphologies, and two-dimensional profiles of the tested samples are shown in Figure 4. As can be observed in Figure 4a, the surface of the control sample (0°) was smoother than the other three, and 2–3 interlaced scratches were distributed in the area shown, with a few bulges around the scratches. Its average surface height fluctuated between −0.47 µm and 0.50 µm. In contrast, the surface morphology of the samples treated by SGP was disorganized. There was a microstructure with plenty of micropits in the surface layer. The densities of micropits in the surface layers of the three samples were similar. The difference was that the average surface height of the sample at a 30° jet angle fluctuated between −1.31 µm and 1.03 µm; when the jet angle was 60°, the average surface height of the sample fluctuated between −1.21 µm and 1.23 µm; and the average surface height of the sample, at a 90° jet angle, fluctuated between −1.61 µm and 1.63 µm. In summary, the flatness of the processed surface worsened as the jet angle of the SGP rose.

The two-dimensional profile of the micropits, the surface roughness R_a_ of the test sample, and the cross-sectional hardness are shown in Figure 5. By comparing with Figure 5a,b, it can be seen that the control (0°) samples had smooth surface profiles. At a jet angle of 30°, there were numerous micropits, with a width and depth of 3.75 µm and 2.17 µm, respectively, on the surface layer of the tested sample. When the jet angle of the SGP was 60°, the width of the micro-pits on the surface of the tested sample decreased to 2.62 µm, while the depth of the micropits increased to 3.51 µm. As the jet angle was further increased to 90°, the width of the micropits was further reduced to 1.94 µm, and the depth of the micropits increased to 4.14 µm.

As shown in Figure 5c, as the jet angle of the SGP increased from 0° to 90°, the roughness of the tested sample surface then increased from 0.48 µm to 1.37 µm with a gradually decreased growing rate. As can be seen in Figure 5d, the cross-section hardness of the control sample (0°) without the SGP treatment fluctuated around 322.4 HV_0.2_. Compared to the control sample (0°), the tested samples treated by SGP exhibited a significant increase in hardness at 10 µm near the surface layer, which was 361.2 HV_0.2_, 374.8 HV_0.2_, and 397.7 HV_0.2_, respectively. Furthermore, its cross-sectional hardness tended to fluctuate and decrease in the depth direction, eventually remaining uniform in the hardness of the matrix material. In conclusion, the surface roughness and surface hardness of the 30CrMnSiA steel samples treated by SGP increased, and the surface roughness and surface hardness of the tested samples increased as the jet angle increased.

The phase composition in the tested surface was investigated by means of X-ray analysis. XRD scanning results of the four samples are shown in Figure 6a. It is clearly shown that the diffraction peaks of the control sample (0°) correspond to martensite (i.e., α′-Fe) and retained austenite (AR). After SGP treatment, the diffraction peak intensity of retained austenite (AR) significantly reduced with increases in the jet angle from 30° to 90°. Moreover, the martensite (α′-Fe) diffraction peaks of samples treated with SGP samples were all offset in the direction of the diffraction angle, and a small amount of Fe_3_O_4_ formed on the processed surface. To further understand the possible mechanism, the grain size and the lattice deformation of these samples were calculated by Williamson–Hall formulas [32], as shown in Figure 6b. The control sample (0°) had a maximum grain size and minimum lattice deformation of 17.38 µm and 0.27, respectively. The SGP treatment resulted in smaller grain sizes and increased lattice deformation in the processed surface. At a jet angle of 30°, the surface grain size of the tested sample was 14.77 µm, and the lattice deformation was 0.28. When the jet angle was increased to 60°, the grain size decreased to 13.31 µm, and the lattice deformation rose to 0.30. The grain size was at a minimum of 12.83 µm, and the lattice deformation was at a maximum of 0.33 at the jet angle of 90°. The martensite in 30CrMnSiA steel was usually in the form of laths, making the surface microstructure denser. Previous studies demonstrated that the dense microstructure with a high lattice deformation and grain refinement could significantly enhance the wear resistance of the material [33]. This evidence further illustrates the benefits that SGP may bring.

### 3.2. Friction Coefficients

Figure 7 shows the typical evolution of friction coefficients concerning jet angle at 0–90° with an interval of 30°. At the initial wear stage, the friction coefficient of samples fluctuated wildly at various jet angles. With the continuous running time of the tester, the friction coefficient of the four samples gradually tended to be stable. The difference is that during the initial wear stage, the friction coefficient of the control sample (0°) was gently between 0.2223 and 0.2379, and the running-in time was 130 s. At a 30° jet angle, the fluctuation of friction coefficient increased further, fluctuating in the range from 0.1978 to 0.2247, with a running-in stage of 286 s. When the jet angle increased to 60°, the friction coefficient of the sample fluctuated sharply in the range from 0.1659 to 0.2007 and a duration of 343 s. The friction coefficient fluctuated most sharply at a jet angle of 90 to a range of 0.1703–0.2044 and continued for 378 s before entering a stable wear stage. It is worth noting that with the jet angle increasing from 0° to 90°, the fluctuation of friction coefficient during the running-in stage became sharp, and the running-in stage extended from 130 s to 378 s.

Figure 8 demonstrates the average friction coefficient for samples with various jet angles at a steady state. There is no doubt that the friction coefficient was significantly related to the jet angle. When the jet angle of the SGP was increased from 0° to 30°, the average friction coefficient of the tested sample decreased rapidly from 0.2235 to 0.1848. As the jet angle continued to be increased from 30° to 60°, the average friction coefficient of the tested sample was further reduced to 0.1609. As the jet angle continued to increase to a maximum of 90°, the average friction coefficient of the tested sample increased from 0.1609 to 0.1756. It can be seen that when the jet angle of the SGP exceeded 60°, the average friction coefficient of the tested sample was increased in the opposite direction, which may have been caused by the excessive surface roughness of the tested sample. It is worth mentioning that the lowest average friction coefficient was obtained at a jet angle of 60°.

### 3.3. Wear Volume and Specific Wear Rate

The wear mark morphologies of the 30CrMnSiA steel samples treated with different SGP jet angles after the wear test are shown in Figure 9 and Figure 10. As the jet angle of the SGP increased from 0° to 30°, the wear degree of the tested sample decreased rapidly, with the average width of the wear marks decreasing from 521.11 µm to 473.33 µm and the average depth of the wear marks decreasing from 5.78 µm to 4.08 µm. When the jet angle continued to rise from 30° to 60°, the width of the wear marks was further reduced to 407.78 µm, and the average depth of the wear mark was reduced to 3.12 µm. As the jet angle increased to 90°, the wear degree of the tested steel increased in reverse, primarily in the width of the wear marks rising from 407.78 µm to 440.01 µm and in the average depth of the wear marks increasing from 3.12 µm to 4.19 µm. In addition, it is shown from Figure 10b that with a jet angle of 30°, the decrease rate of the width from 1.59 µm/° rose to 2.19 µm/°, and the decrease rate of average depth reduced from 0.057 µm/° to 0.032 µm/°. At a jet angle of 60°, the width of the wear mark changed from decreasing at a rate of 2.19 µm/° to increasing at a rate of 1.07 µm/°, and the average depth also changed from decreasing at a rate of 0.032 µm/° to increasing at a rate of 0.032 µm/°.

According to Formulas (1) and (2), the wear volume and specific wear rate of the tested samples versus the SGP injection angle can be obtained, as shown in Figure 11. When the jet angle of SGP increased from 0° to 30°, the wear volume and specific wear rate of the tested sample decreased quickly, from 9.04 × 10^−3^ mm^3^ to 5.79 × 10^−3^ mm^3^ and from 15.13 × 10^−3^ mm^2^/N to 9.66 × 10^−3^ mm^2^/N, respectively. As the jet angle of the SGP increased from 30° to 60°, the wear volume and specific wear rate of the tested sample decreased slowly to 3.82 × 10^−3^ mm^3^ and 6.36 × 10^−3^ mm^2^/N, respectively. As the jet angle continually increased to 90°, the wear volume and specific wear rate increased slowly to 5.53 × 10^−3^ mm^3^ and 9.92 × 10^−3^ mm^2^/N, respectively. With the increase in the jet angle of SGP, the wear volume and specific wear rate of the tested sample all decreased rapidly first, then slowly, and then increased gradually. It is worth noting that when the jet angle was 60°, the wear degree of the tested sample obtained the minimum value, and the wear degree of the tested sample also changed from a decrease to an increase at this angle. It can be concluded that when the jet angle was 60°, the wear mechanism of the tested sample may have changed.

### 3.4. Wear Morphology

As can be seen from Figure 12, the worn areas of all four tested samples showed different degrees of grooves, plastic deformation, fatigue cracks, and fatigue flaking. For the control sample (0°) without being treated by SGP, the grooves of the worn area of the processed surface were the most serious, and there were slight fatigue cracks on the area. In comparison, the grooves in the wear areas of the samples treated by SGP were all lighter than those of the control group. When the jet angle of SGP was 30°, there were serious fatigue cracks and fatigue flaking in the worn area of the tested sample surface. When the jet angle was 60°, there were several shallow grooves distributed in the worn area of the processed surface, and its grooves were shallowest and finest compared with that of the other samples. When the jet angle of SGP was 90°, the worn area showed slight fatigue cracks in addition to shallow grooves in the friction direction.

## 4. Discussion

Previous research has shown that surface microstructures can form sufficient lubricant films on the working surfaces of tribopairs to improve the lubrication properties [34]. During the lubricating process, many micropits formed in the surface layer with SGP treatment can be used as microscale oil pockets for reserving oil molecules, providing an adequate lubricant film for the microareas of friction. As shown in Figure 13, when the wear time from time 1 increased to time 3, the wear degree of the surface layer became more worn, and the microscale oil pockets were also worn. The lubricants stored in the microscale oil pockets acted as a coolant to avoid the increased wear caused by frictional heat. The microscale oil pockets also stored some of the abrasive particles, reducing point contact between the abrasive and the matrix and reducing the wear caused by cold welds.

As is clearly shown in Figure 14, the surface of the control sample (0°) was flat and smooth, while plenty of oil pockets occurred in the surface layer of SGP-treated samples. The oil storage capacity of the oil pockets could be calculated based on the morphology of the micro craters shown above, as shown in Table 3. It can be seen that the microscale oil pockets formed on the surface of the samples with SGP. Furthermore, as the jet angle rose from 30° to 90°, the oil storage capacity of the microscopic oil pockets increased rapidly from 6.37 µm^3^ to 9.16 µm^3^ and then slowly increased to 10.57 µm^3^. In conclusion, the SGP formed plenty of microscopic oil pockets on the sample surface, which were used to store lubricants and enhance the lubricating properties of the surface. Therefore, the wear volume and specific wear rate of the samples treated by SGP were all lower than those of the control sample.

Previous studies have indicated an essential correlation between the friction coefficient of the metal material and its surface contact state [35]. In this work, the average friction coefficient of the sample processed surface exhibited a trend of rapidly decreasing, then slowly decreasing, and then gradually increasing as the jet angle of SGP increased from 0° to 90°. Combined with the cross-sectional morphology of samples and the surface morphology of worn marks, it was revealed that during the running-in stage, the running-in time and the friction fluctuation of the sample all increased when the jet angle increased from 0° to 90°. This was due to the surface roughness of the sample increasing significantly with the rising of the jet angle, which resulted in decreased surface flatness of the sample and less stability during the wear period. Therefore, more time was necessary to attain a steady state. During the steady wear stage, the average friction coefficient of the samples showed a tendency to decrease and then increase with the increase of the jet angle. This was mainly attributed to the many microscopic oil pockets, which improved the lubricating properties of the surface, and which were present on the surface of samples treated by SGP. When the jet angle was 60°, the average friction coefficient was lowest due to the shallowest grooves on the wear area of the processed surface. When the jet angle was 90°, the average coefficient of friction was reversed (increased) due to the excessive surface roughness.

In this study, it can be seen that when the jet angle was 60°, the wear degree of the tested sample was the lowest. The wear degree of the tested sample also shifted from decreased to increase values at this angle, which is likely related to the change of the wear mechanism of the tested sample. By observing and analyzing the worn mark of the tested sample at different jet angles, it can be concluded that when the jet angle of SGP was 0° (control sample), the wear mechanism of the tested sample mainly manifested in severe abrasive wear and slight fatigue wear. During the reciprocal friction process, the low hardness and poor lubricity of the surface layer caused a severe continuous alternating stress on the contact surface of the 30CrMnSiA sample with the Si_3_N_4_ ceramic ball. There was severe dislodging that formed many abrasive particles on the surface microstructure of the tested sample, which resulted in severe abrasive wear. The massive abrasive particles scratching the sample surface also caused the original fatigue flaking area to be replaced by grooves. When the jet angle of SGP was 30°, the wear mechanism of the tested sample was mainly characterized by serious fatigue wear and abrasive wear. The surface hardness of the processed surface, treated with SGP, was significantly increased to 361.2 HV_0.2_, and microscopic oil pockets with an oil storage capacity of 6.37 µm^3^ were formed in the surface layer. When the jet angle reached 60°, the hardness of the processed surface further rose to 374.8 HV_0.2_, and the oil storage capacity of the microscopic oil pockets rapidly expanded to 9.16 µm^3^. The high hardness and excellent lubricating properties caused the sample to manifest slight abrasive wear in the sliding lubrication friction process. When the jet angle was 90°, the oil storage capacity of the microscopic oil pockets formed on the processed surface rose to 10.57 μm^3^, while the surface roughness increased to 1.37 μm. The wear mechanism of the tested sample was characterized by slight fatigue wear and abrasive wear, which may have been attributed to excessive roughness, and resulted in plenty of microscale bulges on the processed surface. During oil lubrication wear, these microscale bulges scratched violently the Si3N4 ball, causing an extensive section of the oil film to crack and the lubrication near the bulge to fail. Moreover, the large amount of friction heat reduced the cooling effect of the oil pockets and eventually led to wear, such as grooves and flaking, which increased the wear of the sample.

In conclusion, the SGP can be used to enhance the lubrication properties of the processed surface of 30CrMnSiA steel samples by forming many micropits with oil storage capacity on the surface layer. There micropits act as microscale lubricating oil pockets. When the jet angle ranged from 0° to 60°, a large number of microscopic oil pockets and the increased hardness were the main reasons for the enhanced wear resistance of the processed surfaces. When the jet angle was 90°, the excessive roughness caused the oil film on the surface to be severely damaged, further leading to a reduction in the lubrication properties of the processed surface and ultimately to the reduced wear resistance of the sample.

The surface microstructures and wear test results at the different jet angles of SGP are shown in Table 4.

## 5. Conclusions

(1)The strengthening grinding process (SGP) is an effective method to enhance the wear resistance of 30CrMnSiA. Due to numerous micropits on the processed surface treated by SGP, the micropits are comparable to microscopic oil pockets that store lubricants during lubrication and friction, which enhance the surface lubrication properties. The SGP also causes grain refinement in the surface layer, raising the surface hardness of the samples.(2)At the conditions of the text experiment, the oil storage capacity of the microscopic oil pockets, the surface hardness, and the surface roughness of the sample are all positively correlated with the jet angle of SGP as the jet angle increases from 0° to 90°, and the running-in time of the tested sample increases, mainly due to the increased roughness of the processed surface. Following steady wear, the friction coefficient, the wear volume, and the specific wear rate of the tested samples demonstrate a rapid decrease, then a slight decrease, and then a slow increase. In addition, the average friction coefficient, wear volume, and wear rate of the samples tested are minimized when the jet angle of the SGP is 60°.(3)In the lubrication and friction test, the wear mechanism of the control sample (0°) is mainly characterized by severe abrasive wear and slight fatigue wear. When the jet angle of the SGP is 30°, the wear mechanism of the tested sample is mainly slight abrasive wear and severe fatigue wear. As the jet angle of the SGP is increased to 60°, the wear mechanism of the tested sample shifts to abrasive wear. At a jet angle of 90°, the wear mechanism shifts back to wear by abrasive and fatigue wear.(4)In this study, when the jet angle ranges from 0° to 60°, a large number of microscopic oil pockets and the increased hardness were the main reasons for the enhanced wear resistance of the processed surfaces. When the jet angle is 90°, the excessive roughness causes the oil film on the surface to be severely damaged, further leading to a reduction in the lubrication properties of the processed surface and ultimately to the reduced wear resistance of the sample.

## Figures and Tables

**Figure 1 materials-15-07380-f001:**
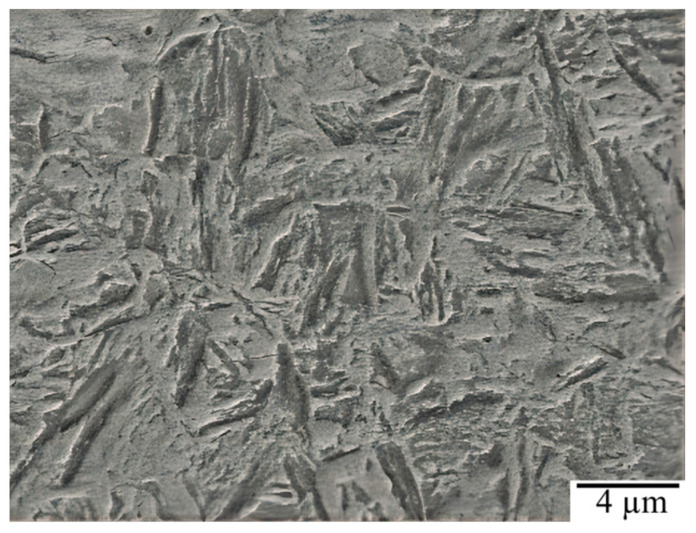
SEM microstructure of 30CrMnSiA.

**Figure 2 materials-15-07380-f002:**
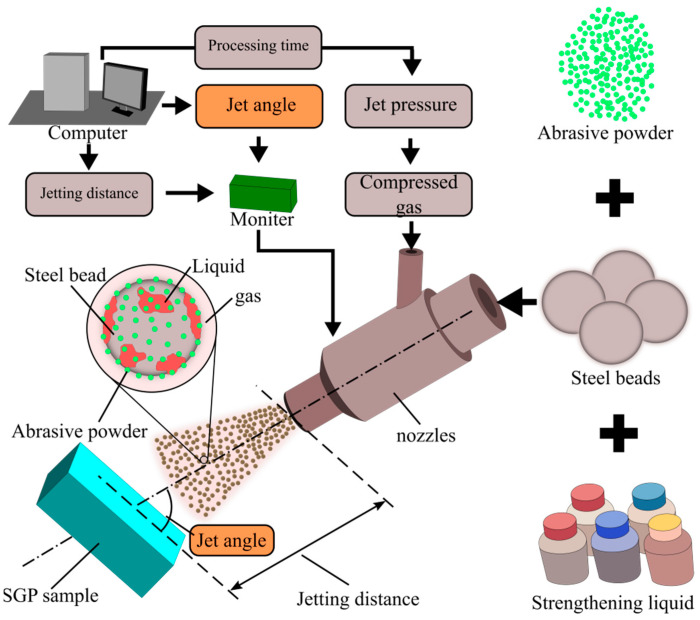
Parameter setting and treatment process of SGP.

**Figure 3 materials-15-07380-f003:**
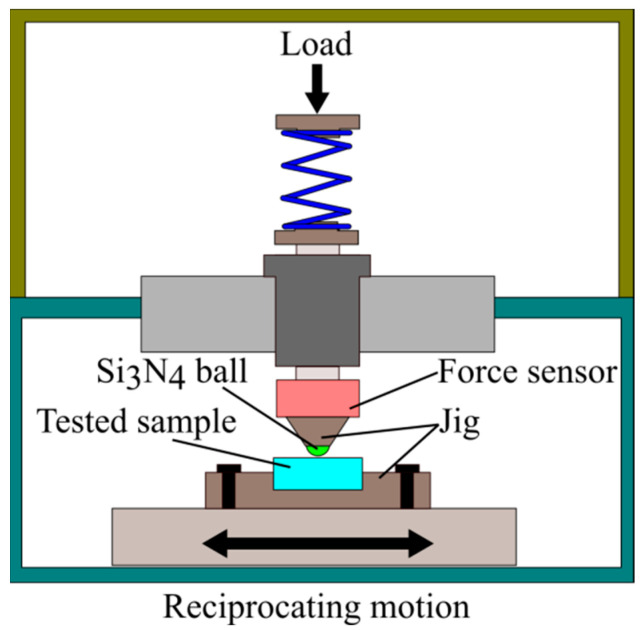
Principle diagram of sliding wear.

**Figure 4 materials-15-07380-f004:**
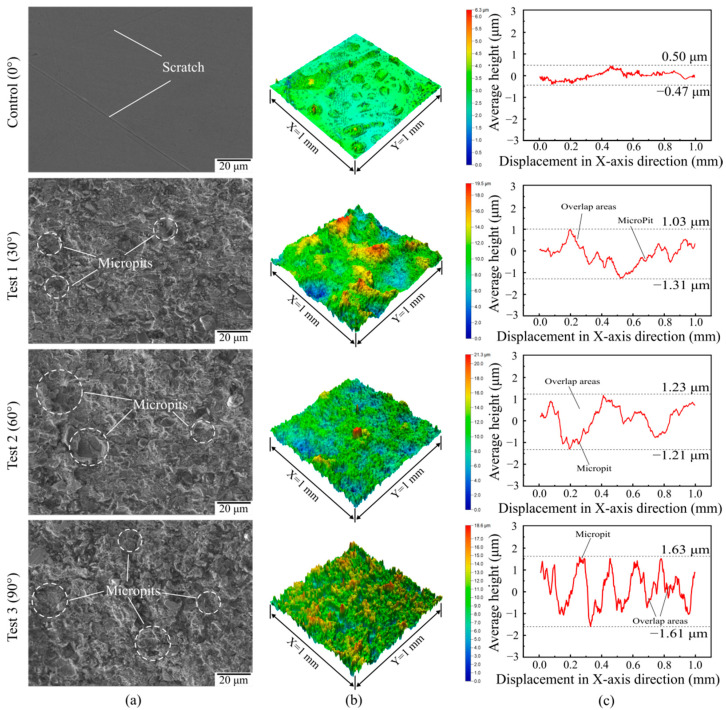
(**a**) SEM maps of surface microstructure versus jet angle; (**b**) three-dimensional morphology at different jet angles; (**c**) two-dimensional profile at different jet angles.

**Figure 5 materials-15-07380-f005:**
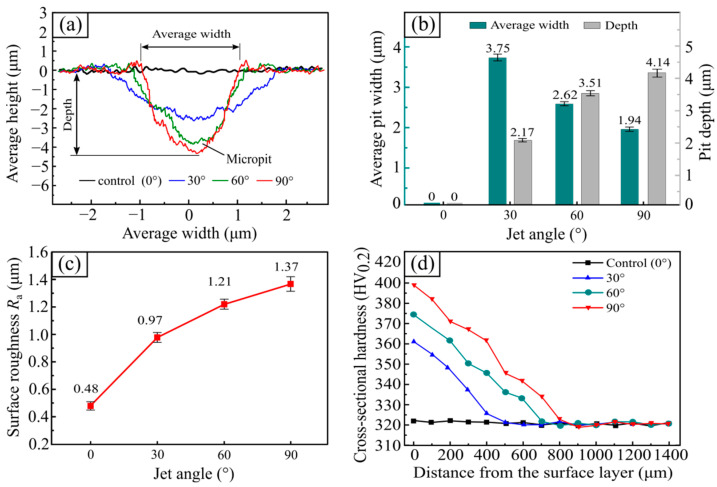
(**a**) Two-dimensional profile pits at different jet angles; (**b**) width and depth of pits versus jet angle; (**c**) surface roughness at different jet angles; (**d**) hardness profiles in the cross-sectional at different jet angles.

**Figure 6 materials-15-07380-f006:**
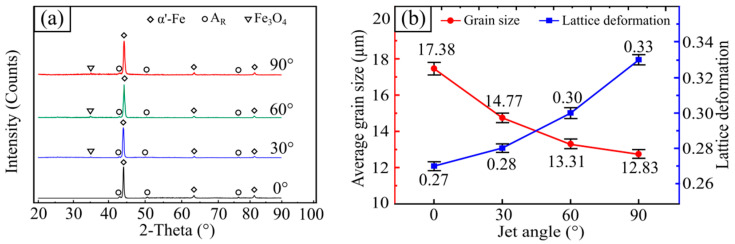
(**a**) The X-ray diffraction pattern characteristics on the tested sample surface; (**b**) the average grain size and lattice deformation at different jet angles.

**Figure 7 materials-15-07380-f007:**
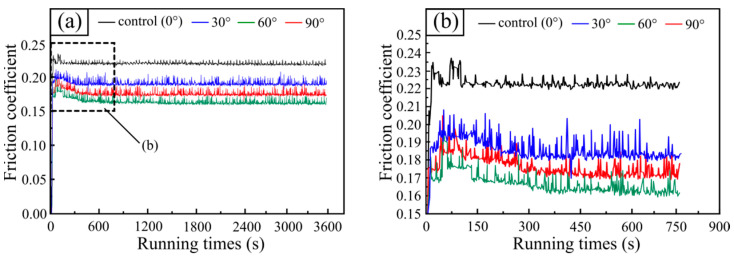
(**a**) Friction coefficients at functions of time at different jet angles; (**b**) friction coefficients during the running-in stage.

**Figure 8 materials-15-07380-f008:**
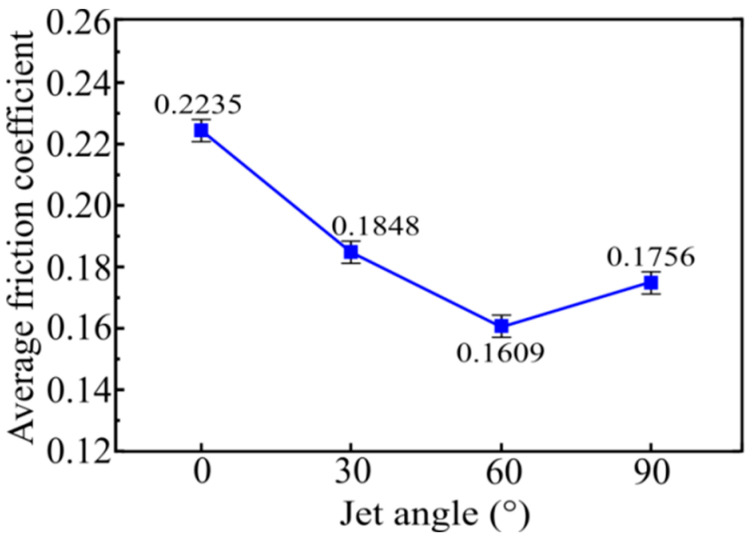
Average friction coefficients at different jet angles.

**Figure 9 materials-15-07380-f009:**
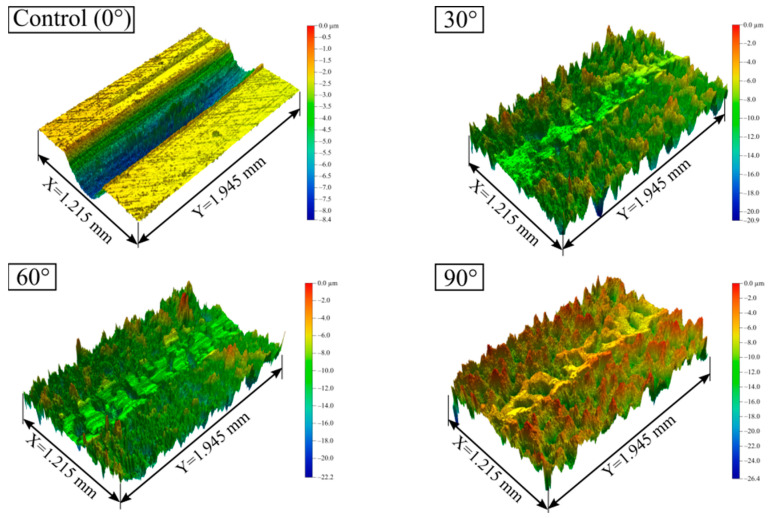
Three-dimensional wear morphology at different jet angles of SGP.

**Figure 10 materials-15-07380-f010:**
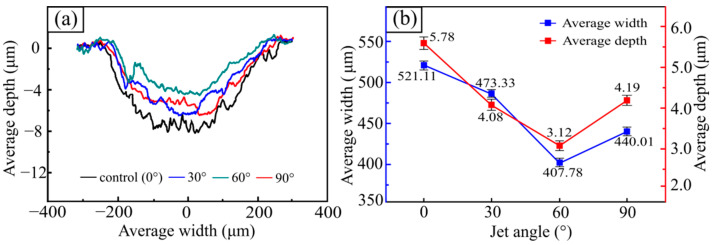
Two-dimensional morphology of wear masks: (**a**) wear mark section profile versus jet angle; (**b**) average wear depth and width versus jet angle.

**Figure 11 materials-15-07380-f011:**
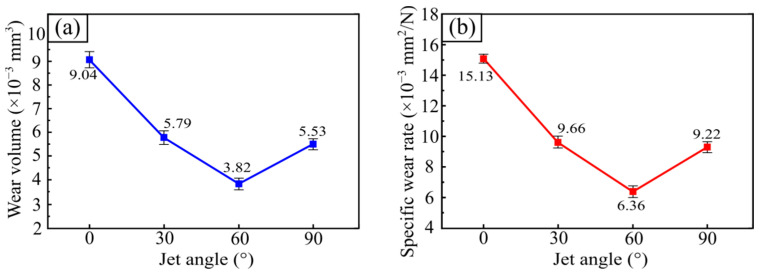
(**a**) The wear volume versus jet angle of SGP; (**b**) specific wear rates at different jet angles of SGP.

**Figure 12 materials-15-07380-f012:**
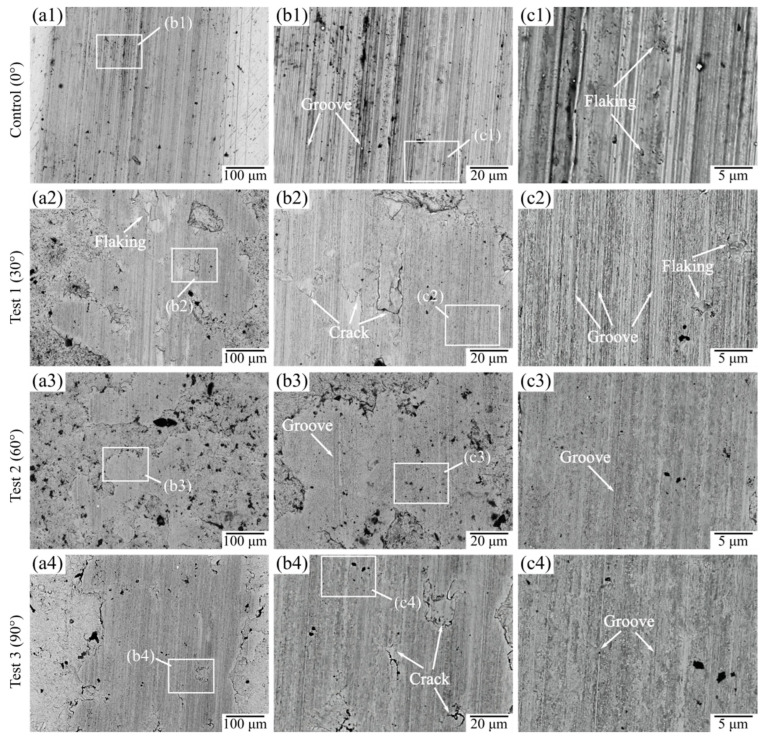
SEM images of worn surface morphology under different jet angles.

**Figure 13 materials-15-07380-f013:**
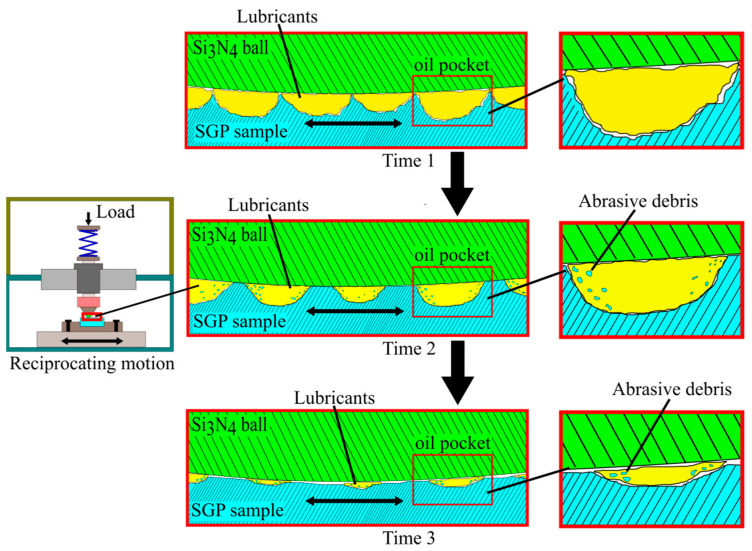
Two-dimensional illustration of the microscope oil pocket.

**Figure 14 materials-15-07380-f014:**
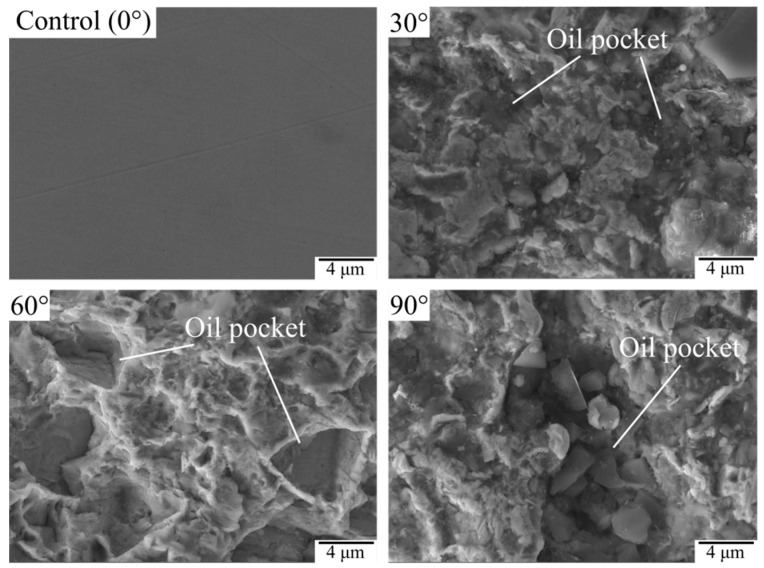
Microscope morphology of the oil pocket.

**Table 1 materials-15-07380-t001:** Chemical composition of 30CrMnSiA Steel (wt. %).

Material	Cr	C	Mn	Si	S	P	Fe
30CrMnSiA	0.98	0.34	0.97	1.03	≤0.01	≤0.01	Bal.

**Table 2 materials-15-07380-t002:** Working parameters of the strengthening grinding process.

Parameters	Control	Test 1	Test 2	Test 3
Jet angle (°)	0	30	60	90
Jet pressure (MPa)	0.7			
Jet distance (mm)	90			
Process time (min)	15			

**Table 3 materials-15-07380-t003:** The oil storage capacity of oil pockets at different jet angles of SGP.

Jet Angle	Control (0°)	30°	60°	90°
Average oil storage volume (µm^3^)	0	12.46	25.35	26.12

**Table 4 materials-15-07380-t004:** Summary of surface microstructures and wear test results versus jet angles.

	Control (0°)	Test 1 (30°)	Test 2 (60°)	Test 3 (90°)
Capacity of oil pockets (μm^3^)	None	12.46	25.35	26.12
Surface hardness (HV_0.2_)	322.4	361.2	374.8	397.7
Surface roughness (μm)	0.48	0.97	1.21	1.37
Running-in time (s)	130	286	343	378
Average friction coefficients	0.2235	0.1848	0.1609	0.1756
Wear volume (×10^−^^3^ mm^3^)	9.04	5.79	3.82	5.53
Specific wear rate (×10^−^^3^ mm^2^/N)	15.13	9.66	6.36	9.22
Slight abrasive wear		✓	✓	✓
Sever abrasive wear	✓			
Slight fatigue wear	✓			✓
Sever fatigue wear		✓		

The wear mechanical of tested samples is marked with a “✓”.

## Data Availability

Not applicable.

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
