# Peer review of "The Tribological Properties of 30CrMnSiA Bearing Steels Treated by the Strengthening Grinding Process under Lubrication Wear"

_materials, 2022, doi:10.3390/ma15207380_

Round 1
Reviewer 1 Report
I have carefully read the article titled “The Tribological Properties of 30CrMnSiA bearing steels treated by Strengthening Grinding Process under Lubrication Wear” that you sent in the first evaluation. When I evaluated the article in terms of content and format, I realized it was high quality. That's why I suggested a minor revision without much comment. Additional comments I made about the article are given below.
1. I think the work is original. There are many studies in the literature about 30CrMnSiA bearing steels. However, the authors aim to improve the surface properties with a different method.
2. The article is well organized. Studies in the literature were compared, detailed analyzes were made using different methods, and satisfactory results were obtained.
3. All chapters are written in a very legible and understandable way.
4. The authors presented their experiments and results transparently. I think the results are consistent with the evidence and arguments presented. The analysis results provide successful and high-resolution figures. According to the comments made in the article, the main question is clearly addressed.
Author Response
Response to Reviewer 1 Comments
Point 1: I think the work is original. There are many studies in the literature about 30CrMnSiA bearing steels. However, the authors aim to improve the surface properties with a different method.
Response 1: The article uses the strengthening grinding process to enhance the surface properties of 30CrMnSiA bearing steel; I will continue to optimize the article; thanks.
Point 2: The article is well organized. Studies in the literature were compared, detailed analyzes were made using different methods, and satisfactory results were obtained.
Response 2: Thank you for your approval, and I will continue to improve this article.
Point 3: All chapters are written in a very legible and understandable way.
Response 3: I will enhance the description of the research background and methods. Thank you.
Point 4: The authors presented their experiments and results transparently. I think the results are consistent with the evidence and arguments presented. The analysis results provide successful and high-resolution figures. According to the comments made in the article, the main question is clearly addressed.
Response 4: Thank you for your approval, and I will continue to improve the descriptions of the chapters of the results and the discussion.
Reviewer 2 Report
After reviewing the manuscript, I think that the subject is relevant and interesting. Providing the surface of bearing steel by infinite numbers of oil pockets that guarantee a continuous oil film on the sliding surface of bearing steel is original experimental finding and adds solutions for the lubrication of rolling bearing in starved sliding condition. It is well known that failure of most of rolling bearings is due to the marginal lubrication. So, the subject of the paper is promising. The language of the paper is clear and easy to read followed by consistent conclusions.
Therefore, I strongly recommend the publication of the paper.
Author Response
Point 1: After reviewing the manuscript, I think that the subject is relevant and interesting. Providing the surface of bearing steel by infinite numbers of oil pockets that guarantee a continuous oil film on the sliding surface of bearing steel is original experimental finding and adds solutions for the lubrication of rolling bearing in starved sliding condition. It is well known that failure of most of rolling bearings is due to the marginal lubrication. So, the subject of the paper is promising. The language of the paper is clear and easy to read followed by consistent conclusions.
Therefore, I strongly recommend the publication of the paper.
Response 1: Thank you very much for your approval of my paper and I will further improve the expression of English in my article.
Reviewer 3 Report
In the conditions of modern engineering, production mechanisms, household appliances, appliances and vehicles are increasingly equipped with bearings. Such a comprehensive demand causes tougher requirements for their service properties. This, in turn, determines more stringent criteria for selecting materials for the manufacture of bearings and strictly regulates the properties that bearing steel should have, because despite the differences in design, these bearing elements operate under conditions of high local loads, intense vibrations and high speeds. Therefore, the results of the article are very relevant for science and practical use. But there are some questions:
To ensure high performance and reliability of bearings, steels from which rolling elements and rings are made must have:
1. What are the mechanical properties of 30CrMnSiA steel?
2. What is known about contact fatigue and crack resistance of 30CrMnSiA steel?
3. A ring bearing drawing would be very helpful. After all, is it necessary that the steel provides dimensional stability during operation?
4. Which roughness parameter was investigated and shown in Fig. 5s. More than 20 surface roughness parameters are known. We ask the authors to indicate which was used?
5. On fig. 13 is a schematic illustration of the microscope oil pocket, but it might also be worth showing the surface quality using by the Abbott-Firestone curve. Professor Dzyura's article https://www.mdpi.com/2075-1702/9/6/116 shows a similar approach. I think it will be useful for you.
Author Response
Response to Reviewer 3 Comments
Point 1: What are the mechanical properties of 30CrMnSiA steel?
Response 1: For 30CrMnSiA bearing steel, the mechanical properties include elasticity, plasticity, stiffness, hardness, wear resistance, impact resistance, fatigue resistance, and so on, which are the ability to resist load damage during operation. This paper mainly focuses on wear resistance and tries to promote wear resistance through surface-strengthening technology.
Point 2: What is known about contact fatigue and crack resistance of 30CrMnSiA steel?
Response 2: 30CrMnSiA bearing steel is mainly sent to rolling contact fatigue, and the mechanism can be divided into surface-induced and subsurface-induced. The former will cause surface corrosion, and the latter will cause fatigue spalling of the surface layer material, and there is no way to avoid this. The subsurface-induced fatigue release is the main cause of fatigue crack expansion in 30CrMnSiA bearing steel.
Point 3: A ring bearing drawing would be very helpful. After all, is it necessary that the steel provides dimensional stability during operation?
Response 3: I appreciate your advice, but the research of the article is aimed at the wear resistance of bearing materials and is not yet applied to the engineering field. I will continue to improve the quality of the images and the presentation
Point 4: Which roughness parameter was investigated and shown in Fig. 5s. More than 20 surface roughness parameters are known. We ask the authors to indicate which was used?
Response 4: The surface roughness illustrated in Figure 5 is the arithmetic average deviation from the absolute value of the surface profile offset (Ra), and I will add to the pictures and text.
Point 5: On fig. 13 is a schematic illustration of the microscope oil pocket, but it might also be worth showing the surface quality using by the Abbott-Firestone curve. Professor Dzyura's article https://www.mdpi.com/2075-1702/9/6/116 shows a similar approach. I think it will be useful for you.
Response 5: Thank you for your suggestions. I got a great benefit from reading Professor Dzyura's article, especially the improvement of oil film lubrication by microrelief, which I will also quote. I think the way of characterization in it is worthy of my study. Figure 13 is mainly to show the role of the oil pockets in the wear process, so I think it is better to make it in the format of a procedure.
Round 2
Reviewer 3 Report
Accept.